# Exploring Chain of Thought Style Prompting for Text-to-SQL

**Chang-You Tai**[*], **Ziru Chen**[*†], **Tianshu Zhang, Xiang Deng, Huan Sun**[†]

The Ohio State University

{tai.97, chen.8336, zhang.11535, deng.595, sun.397}@osu.edu

## Abstract

In-context learning with large language models (LLMs) has recently caught increasing attention due to its superior few-shot performance on various tasks. However, its performance on text-to-SQL parsing still has much room for improvement. In this paper, we hypothesize that a crucial aspect of LLMs to improve for text-to-SQL parsing is their multi-step reasoning ability. Thus, we systematically study how to enhance LLMs' reasoning ability through chain of thought (CoT) style prompting, including the original chain-of-thought prompting (Wei et al., 2022b) and least-to-most prompting (Zhou et al., 2023). Our experiments demonstrate that iterative prompting as in Zhou et al. (2023) may be unnecessary for text-to-SQL parsing, and using detailed reasoning steps tends to have more error propagation issues. Based on these findings, we propose a new CoT-style prompting method for text-to-SQL parsing. It brings 5.2 and 6.5 point absolute gains on the Spider development set and the Spider Realistic set, respectively, compared to the standard prompting method without reasoning steps; 2.4 and 1.5 point absolute gains, compared to the least-to-most prompting method[1].

## 1 Introduction

Text-to-SQL parsing, the task of mapping a natural language utterance to a SQL query, has found wide applications in building language agents for databases and piqued significant research interest in recent years (Deng et al., 2021; Yu et al., 2021; Rajkumar et al., 2022; Hongjin et al., 2023; Ni et al., 2023). To develop a text-to-SQL parser, a prevalent approach is to collect labeled data and train a model via supervised learning (Shaw et al., 2021; Scholak et al., 2021). While effective, this approach necessitates a considerable amount of training data, which

is costly to obtain because annotating SQL queries requires programming expertise.

As an alternative to supervised learning, in-context learning (Brown et al., 2020), an emergent capability of large language models (LLMs), alleviates the need for large-scale data. With only a few examples, in-context learning enables LLMs to demonstrate performance comparable to or even better than fully supervised models on many NLP tasks, such as question answering, machine translation, and natural language inference (Chowdhery et al., 2022; Kojima et al., 2022; Wei et al., 2022b,a; Brohan et al., 2023). When applied to text-to-SQL parsing, in-context learning has also shown encouraging results (Rajkumar et al., 2022; Chang et al., 2023b; Liu et al., 2023a), but there is still much room for improvement.

We hypothesize that a crucial aspect of LLMs to improve for text-to-SQL parsing is their multi-step reasoning ability. Even for a seemingly simple question, such as "What is the ID of Kyle," a model has to ground it to the given database schema, infer the relational algebra among schema items, and construct syntactically correct SQL clauses. To enhance LLMs' reasoning capabilities, chain of thought (CoT) style prompting methods (Wei et al., 2022b; Zhou et al., 2023) are proposed and have shown promising results. However, how to apply CoT-style prompting to text-to-SQL parsing remains under-explored, and we fill this gap by systematically exploring CoT-style prompting for text-to-SQL parsing. Specifically, we seek to answer two research questions: (*RQ1*) Which prompting style is better, generating all reasoning steps in one pass, or iterative prompting and problem solving? (*RQ2*) Do more detailed reasoning steps lead to better results for text-to-SQL parsing?

To address these questions, we adapt two widely used prompting methods for text-to-SQL parsing. As the first method, we apply *chain-of-thought prompting* (Wei et al., 2022b) by drawing an anal-

---

[*]Equal Contribution.
[†]Corresponding authors.
[1]We include all our prompts in the appendix.

Figure 1: Example model input and expected outputs for four CoT style prompting methods applied to text-to-SQL parsing: A. Chain-of-Thought, B. Least-to-Most, C. QDecomp, and D. QDecomp + InterCOL, where C and D are our proposed methods.

ogy between its problem-solving process and the execution procedure of a SQL query (Figure 1A). Referring to the logical execution order of SQL clauses (Narechania et al., 2021), we compose the intermediate execution steps in natural language and prompt LLMs to derive them before generating the SQL query. For the second method, we follow Zhou et al. (2023) to apply *least-to-most prompting* in two stages: (1) problem reduction: generate a series of sub-questions from the original question and (2) problem solving: iteratively translate each sub-question into its corresponding SQL query, with the original question as the last sub-question, as shown in Figure 1B. With a careful analysis (Section 5.2), we find that directly applying these two methods for text-to-SQL parsing tends to introduce error propagation issues frequently. Also, the iterative process in least-to-most prompting incurs more computational costs to generate each SQL query.

Therefore, we propose a new CoT-style prompting method called *question-decomposition prompting* (QDecomp, Figure 1C). Similar to chain-of-thought prompting, QDecomp generates a sequence of reasoning steps followed by the natural language question in one pass. Instead of generating the intermediate execution steps, we instruct LLMs to decompose the original complex question, akin to the problem reduction stage in least-to-most prompting. Furthermore, to help LLMs ground database schemas, we design a variant of question decomposition prompting (QDecomp+InterCOL, Figure 1D) by incrementally including the table and column names involved in each sub-question.

We conduct comprehensive evaluations on two cross-domain text-to-SQL datasets, Spider (Yu et al., 2018) and Spider Realistic (Deng et al., 2021). Compared to the standard prompting method without reasoning steps, QDecomp + InterCOL brings 5.2 and 6.5 point absolute gains on the Spider development set and the Spider Realistic set, respectively. It also brings 2.4 and 1.5 point absolute gains compared to least-to-most prompting. Our results suggest that it may be unnecessary to perform iterative prompting, which is also computationally costly (*RQ1*). Besides, our analysis shows that our QDecomp+InterCOL method reduces the chance of error propagation by providing less detailed reasoning steps and generating

the SQL query in one pass (*RQ2*). Meanwhile, it includes key schema information in reasoning, which is still beneficial to database grounding. Further, we evaluate the robustness of our proposed methods by varying the number, selection, and format of in-context examples, providing useful guidelines for designing text-to-SQL prompting strategies. We also extend our evaluation to three single-domain datasets (Zelle and Mooney, 1996; Iyer et al., 2017; Yaghmazadeh et al., 2017) and show our proposed method can achieve strong performance consistently across different datasets.

## 2 Related Work

**LLM and CoT-Style Prompting.** As large language models (LLMs) advance (Brown et al., 2020; Chowdhery et al., 2022), in-context learning emerged as a new paradigm in natural language processing (Liu et al., 2023b). Although LLMs can achieve outstanding performance by prompting them with few-shot examples in context, they struggle with tasks that require multi-step reasoning. As a solution, Wei et al. (2022b) proposed chain-of-thought prompting. By explicitly describing intermediate reasoning steps to answer a complex question in the prompts, chain-of-thought prompting improves the accuracy of LLMs by a large margin across many natural language reasoning tasks. Besides, Zhou et al. (2023) proposed least-to-most prompting to solve complex problems in two stages. The method first prompts LLMs to generate a list of sub-questions as a decomposition of the given problem. Then, it uses the sub-questions to guide LLMs to incrementally solve each of them and derive a correct final answer. However, how to apply these two CoT-style prompting methods to text-to-SQL parsing remains under-explored.

We fill this gap by systematically exploring several CoT-style prompting methods for the task. In particular, we propose a new CoT-style prompting method that guides LLMs to perform reasoning via question decomposition. Question decomposition is a method that converts a complex problem into a sequence of simpler sub-questions (Gupta and Lewis, 2018; Min et al., 2019). Our work refers to existing question decomposition methods for text-to-SQL parsing (Wolfson et al., 2020, 2022) and presents a novel CoT-style prompting method to improve LLMs' performance. We conduct comprehensive experiments and show that our question decomposition prompting outperforms chain-of-thought prompting and least-to-most prompting on several text-to-SQL datasets. Our experiments validate our hypothesis that text-to-SQL parsing indeed requires multi-step reasoning, and carefully designed CoT-style prompting can help LLMs achieve higher parsing accuracy.

**Text-to-SQL Semantic Parsing.** Text-to-SQL semantic parsing has long been studied to build language agents for database applications (Dahl et al., 1994; Zelle and Mooney, 1996). Since the release of Spider (Yu et al., 2018), a cross-database text-to-SQL benchmark, many parsers have been developed on top of language models to better understand various database schemas (Wang et al., 2020; Yu et al., 2021; Deng et al., 2021). Recent work starts to explore the potential of LLMs, such as Codex (Chen et al., 2021), in text-to-SQL parsing by including database schemas in the prompts (Rajkumar et al., 2022) or retrieving similar questions as few-shot examples (Hongjin et al., 2023). Orthogonal to these methods, our question decomposition prompting teaches LLM to perform multi-step reasoning for text-to-SQL parsing without additional engineering efforts. With a few in-context examples, an LLM, such as Codex in our experiments, can learn to decompose natural language questions and predict table and column names (Section 3) incrementally in each step.

Our method demonstrates comparable performance to RASAT+PICARD (Qi et al., 2022), a fine-tuned text-to-SQL parser, on the Spider development set without using relational structures or constrained decoding. Compared to other LLM-based methods, it achieves better execution accuracy than DIN-SQL (Pourreza and Rafiei, 2023) on the Spider development set in a single pass, while DIN-SQL requires iterative prompting. Although our method shows lower execution accuracy than LEVER (Ni et al., 2023), we note that LEVER's verifier model is fine-tuned on the full Spider training set, which may have extra advantages over our method. Also, LEVER uses the execution results of SQL queries, which provides extra information for better database grounding. We leave the incorporation of database contents beyond table and column names into our method as future work.

## 3 Prompting for Multi-Step Reasoning in Text-to-SQL

In this section, we outline three CoT-style prompting methods that teach an LLM to perform

multi-step reasoning. We first describe how we adapt chain-of-thought and least-to-most prompting for text-to-SQL parsing. Then, we propose a novel prompting method, question decomposition prompting (QDecomp), and its variant QDecomp+InterCOL. Figure 1 demonstrates different prompting methods, and we provide more examples in Appendix A. For all experiments, we use Codex (Chen et al., 2021), `code-davinci-002`, as the LLM. The experiments were conducted between January and March 2023 through OpenAI API, using greedy decoding with temperature 0.

## 3.1 Chain-of-Thought Prompting

Chain-of-thought prompting (Wei et al., 2022b) aims to improve LLMs' reasoning ability by generating a series of intermediate steps before predicting the final answer. For text-to-SQL parsing, one challenge is how to come up with the reasoning steps to predict the SQL query (i.e., final answer in our case). In our work, we use each clause in the SQL query to compose a reasoning step in CoT prompting. Specifically, inspired by Narechania et al. (2021), we use natural language templates to describe each SQL clause and chain them in the logical execution order of the SQL query. For example, the logical execution order for the SQL query in Figure 1A is first the `FROM` clause, then the `WHERE` clause, and finally the `SELECT` clause. Following this order, we assemble the natural language description of each clause in the query to compose its CoT reasoning steps.

## 3.2 Least-to-Most Prompting

Unlike chain-of-thought prompting, which instructs LLMs to generate all reasoning steps in a single pass, least-to-most prompting (Zhou et al., 2023) tackles complex questions by prompting LLMs in two stages: problem reduction and problem solving. During problem reduction, it prompts the LLM to generate a series of sub-questions from the original complex question. During problem solving, it prompts the LLM with one sub-question at a time and iteratively builds up the final solution.

To derive the sub-questions for problem reduction, we segment the original question following three principles: (1) If the question has multiple sentences, we treat each sentence as a sub-question. (2) We further decompose each sentence by conjunction words (such as "and," "or," and "but") and prepositions (such as "for," "with," and "without").

(3) For each decomposition, we remove words and phrases that may leak the information in any subsequent questions. This segmentation allows the LLM to focus on parsing each sub-question, thereby decreasing the complexity of the original problem (Wolfson et al., 2022).

For instance, the question "Show first name, last name, age for all female students. Their sex is F." in Figure 1B would derive two sub-questions: (a) "Show first name, last name, age for all students." (b) "Show first name, last name, age for all female students. Their sex is F." This decomposition follows principle (1) and (3) by removing the second sentence and the information-leaking word "female" from the original question to construct the first step. For the first sub-question, the LLM only needs to construct the `SELECT` and `FROM` clauses. Then for the second sub-question, the LLM can build upon the SQL query generated for the first sub-question, and focus solely on the `WHERE` clause.

## 3.3 Question Decomposition Prompting

We propose a new prompting method, question decomposition prompting (QDecomp). Similar to chain-of-thought, QDecomp generates intermediate reasoning steps and the final SQL query in a single pass. Instead of using the logical execution procedure of SQL as in CoT, we follow the problem reduction stage in least-to-most prompting and instruct the LLM to decompose the original complex question as the reasoning steps. Through this design, we hope to explore (1) the potential advantage of using question decomposition over the logical execution procedure of SQL clauses for composing reasoning steps; (2) whether an iterative process as in least-to-most prompting is necessary.

On top of that, we propose a variant, QDecomp+InterCOL, to alleviate the well-known table/column linking issue in text-to-SQL parsing (Wang et al., 2020). Specifically, we augment the in-context examples to prompt the LLM to identify any corresponding table/column names when generating each sub-question. Given a sub-question and its corresponding SQL parse, we annotate all table-column pairs mentioned in the parse as ground-truth. For star operators (*), we sample a random column from tables mentioned in the same (sub-)query. If a table-column pair has been mentioned in the SQL parse of a sub-question, we would exclude it from the annotations of all subsequent steps. If a sub-question does not have

any table-column pairs to annotate, we randomly choose one pair from preceding steps.

We include examples of these two methods in Figure 1C and 1D. Following the same decomposition method in least-to-most prompting, the example has two sub-questions. In Figure 1D, for the first sub-question, "Show first name, last name, age for all students," we expect the model to highlight the table "student" and its columns "fname," "lname," and "age," as they appear in the SQL parse of this sub-question. Then, for the follow-up question, the model is expected to identify the table "student" and its column "sex," which is not mentioned in the previous step.

In addition to the prompting methods mentioned above, we also include **the standard prompting method** as the baseline in our experiments. It uses question-SQL pairs as in-context examples to prompt LLMs to directly parse a natural language question to its corresponding SQL query without generating any intermediate reasoning step.

## 4 Experimental Setup

### 4.1 Datasets

**Spider (Yu et al., 2018).** Spider is a commonly used benchmark to evaluate text-to-SQL parsing in a cross-database setting, which requires models to generalize to novel database schemas. The dataset consists of 7,000 question-query pairs in the training set and 1,034 pairs in the development set, covering 200 different databases and 138 domains. In this paper, due to the unavailability of the test set, we evaluate on the Spider development set to demonstrate the effectiveness of our question decomposition prompting methods.

**Spider Realistic (Deng et al., 2021).** Spider Realistic is a more challenging version of the Spider development set. It modifies the natural language questions in Spider by removing or paraphrasing explicit mentions of column names to generate a more realistic dataset that reflects real-world scenarios, where questions rarely contain explicit mentions of column names. The final dataset comprises a total of 508 question-query pairs.

### 4.2 In-context Example Selection

To show the robustness of question decomposition prompting, we consider two ways of choosing in-context examples: **random selection** and

**difficulty-based selection**. In our main results, we use random selection for its simplicity and ease of replication. Additionally, in Section 5.3, we compare results obtained using random selection with those obtained using difficulty-based selection.

For **random selection**, we uniformly sample in-context examples from the Spider training set at random. For **difficulty-based selection**, we first group the Spider training examples into four difficulty levels, pre-defined by Yu et al. (2018), including easy, medium, hard, and extra-hard. Then, we devise three methods to randomly select in-context examples based on their difficulties: (G1) sampling an equal number of examples at each difficulty level, (G2) sampling the same number of examples from the hard level and the extra-hard level respectively, and (G3) sample all examples from the extra-hard level.

### 4.3 Prompt Formats

We also experiment with two prompt formats introduced by Rajkumar et al. (2022), **API Docs** and **Create Table + Select 3**. Both formats have their own advantages and can be utilized together with any prompting method in Section 3.

**API Docs** format represents database schemas as Python API comments, which only includes the table and column names. This format reduces the prompt length for each example, so we may include more in-context demonstrations from databases in different domains to increase diversity. In comparison, **Create Table + Select 3** format adheres more closely to the SQLite standards, but with much longer prompts[2]. It represents a database schema using the `CREATE TABLE` command, which provides more information, such as column data types and foreign key declaration. Besides, this format includes the results of executing `SELECT * FROM T LIMIT 3` for each table `T` in the database as SQL comments. In Section 5.3, we show that API Docs format can achieve competitive performance compared to the Create Table + Select 3 format. Thus, we primarily use the API Docs format in our experiments due to its efficiency.

### 4.4 Evaluation Metric

We use test-suite execution accuracy (Zhong et al., 2020) to evaluate different prompting methods, in-context example selection strategies, and prompt

---

[2]https://platform.openai.com/examples/default-sql-translate

| Method | Spider Dev | | | | | Spider Realistic |
| --- | --- | --- | --- | --- | --- | --- |
| | Easy | Medium | Hard | Extra Hard | Overall TS (Overall EX) | Overall TS (Overall EX) |
| Standard | 86.8 | 65.3 | 50.3 | 36.0 | $63.2 \pm 2.51$ ($68.7 \pm 4.08$) | $51.0 \pm 4.29$ ($62.5 \pm 4.01$) |
| Chain-of-Thought | 73.9 | 64.5 | 44.6 | 23.4 | $56.8 \pm 5.83$ ($53.9 \pm 7.21$) | $50.3 \pm 4.94$ ($53.4 \pm 9.19$) |
| Least-to-Most | 88.1 | 68.7 | 52.9 | 39.5 | $66.0 \pm 2.48$ ($68.9 \pm 3.44$) | $55.0 \pm 2.51$ ($\underline{63.3} \pm 2.73$) |
| Least-to-Most (G3) | 80.3 | 64.6 | 52.8 | $\underline{45.3}$ | $63.3 \pm 1.95$ ($\underline{73.8} \pm 1.72$) | -* |
| QDecomp | **89.8** | $\underline{71.3}$ | $\underline{53.1}$ | 38.6 | $67.4 \pm 1.89$ ($70.7 \pm 2.80$) | $\underline{55.8} \pm 2.01$ (**$65.8 \pm 2.29$**) |
| + InterCOL | $\underline{89.6}$ | **74.1** | 52.4 | 38.1 | $\underline{68.4} \pm 2.05$ ($69.7 \pm 5.82$) | **$56.5 \pm 2.05$** ($\underline{63.3} \pm 4.19$) |
| + InterCOL (G3) | 88.7 | 71.1 | **56.8** | **45.7** | **$68.8 \pm 1.16$** (**$78.2 \pm 1.07$**) | -* |

Table 1: 8-shot test-suite (TS) accuracy of Codex on Spider Dev and Spider Realistic using different prompting methods and API Doc format. In-context examples are randomly selected except for the two rows marked with G3, where we only use extra-hard SQL queries (Section 4.2). We also include the overall standard execution accuracy (EX) in parenthesis for reference. For each method, we repeat the experiments with 5 different sets of in-context examples and report the average performances with their standard deviation. *We were not able to run G3 example selection on Spider Realistic before Codex became unavailable.

| | SELECT | WHERE | GROUP BY | ORDER BY | KEYWORDS |
| --- | --- | --- | --- | --- | --- |
| Standard | 89.8 | 66.1 | 74.7 | 83.0 | 84.2 |
| Chain-of-Thought | 83.5 | 70.7 | 67.1 | 72.8 | 76.9 |
| Least-to-Most | 90.0 | 70.7 | 72.5 | 82.4 | 84.3 |
| QDecomp | 91.2 | 70.7 | **77.2** | 85.1 | **86.4** |
| + InterCOL | **91.4** | **72.4** | 76.6 | **85.3** | 86.0 |

Table 2: 8-shot component matching accuracy of Codex on the Spider development set.

formats. Leveraging the idea of "code coverage" in software testing (Miller and Maloney, 1963), the metric synthesizes a large number of databases as "test cases" and compares the execution results of the predicted and gold SQL queries on all of them. In this way, test-suite accuracy reduces the number of false positives (i.e., semantically different SQL queries that happen to have the same execution result) in standard execution accuracy, which compares execution results on only one database. As shown in Table 1, standard prompting's test-suite accuracy falls behind least-to-most prompting. However, their standard execution accuracy results are very close, which might be misleading.

## 5 Results and Analysis

By analyzing our experimental results, we seek to answer the following two research questions:

- *RQ1*: Which prompting style is better, generating all reasoning steps in one pass, or iterative prompting and problem solving?

- *RQ2*: Do more detailed reasoning steps lead to better results for text-to-SQL parsing?

### 5.1 Main Results

Through comprehensive experiments on Spider Dev and Spider Realistic (Table 1), we show that our proposed question decomposition (QDecomp)

prompting and its variant (QDecomp+InterCOL) consistently outperform two existing methods, chain-of-thought and least-to-most prompting. Specifically, QDecomp+InterCOL achieves 68.4% test-suite accuracy on the Spider development set and 56.5% on the Spider Realistic set. Compared to the standard prompting, it brings 5.2% and 6.5% point absolute gains, respectively. Compared to least-to-most prompting (the second best method), it brings 2.4% and 1.5% point absolute gains. Furthermore, when using extra-hard (G3) in-context examples, we can improve the execution accuracy of QDecomp+InterCOL prompting to 78.2%, which is comparable to RASAT+PICARD (Qi et al., 2022), a strong fine-tuned text-to-SQL parser. In contrast, least-to-most prompting does not gain too much execution accuracy (73.8%) from G3 in-context examples and even has decreased test-suite accuracy (63.3%). We will present more analysis on this contrast in Section 5.3.

Additionally, the experiments show that iteratively solving a series of sub-questions may not be necessary for text-to-SQL parsing (*RQ1*). Although chain-of-thought prompting (56.8%) underperforms least-to-most prompting (66.0%) on the Spider development set, these two methods have several distinct designs other than iterative prompting, so we cannot directly answer *RQ1* by comparing them. With our QDecomp prompting,

| Question | Which dogs are of the rarest breed? Show their names and treatment dates. |
|---|---|
| Chain-of-Thought | # This query chooses records from the dogs table, followed by joining the breeds table on the breed_code column. It then joins the treatments table on the dog_id column. It then groups the results by breed_name. It then orders the results by COUNT(DISTINCT T1.dog_id) in descending order. It then selects the breed_name and date_of_treatment columns.
# Thus, the answer for the question is: Which dogs are of the rarest breed? Show their names and treatment dates. `SELECT T2.breed_name , T3.date_of_treatment FROM dogs AS T1 JOIN breeds AS T2 ON T1.breed_code = T2.breed_code JOIN treatments AS T3 ON T1.dog_id = T3.dog_id GROUP BY T2.breed_name ORDER BY COUNT(DISTINCT T1.dog_id) DESC` |
| Gold | `SELECT T1.name, T2.date_of_treatment FROM dogs AS T1 JOIN Treatments AS T2 ON T1.dog_id = T2.dog_id WHERE T1.breed_code = (SELECT breed_code FROM Dogs GROUP BY breed_code ORDER BY count(*) ASC LIMIT 1)` |

Table 3: An example error made by Codex with chain-of-thought prompting.

| Question | How many United Airlines flights go to City 'Aberdeen'? |
|---|---|
| Least-to-Most | Q: How many United Airlines flights go to City?
A:`SELECT count(*) FROM flights WHERE airline = 'United Airlines' AND destairport = 'City'`
Q: How many United Airlines flights go to City 'Aberdeen'?
A:`SELECT count(*) FROM flights WHERE airline = 'United Airlines' AND destairport = 'Aberdeen'` |
| Gold | Q: How many United Airlines flights?
A:`SELECT count(*) FROM flights AS T1 JOIN airlines AS T2 ON T2.uid = T1.airline WHERE T2.airline = 'United Airlines'`
Q: How many United Airlines flights go to City 'Aberdeen'?
A:`SELECT count(*) FROM flights AS T1 JOIN airports AS T2 ON T1.destairport = T2.airportcode JOIN airlines AS T3 ON T3.uid = T1.airline WHERE T2.city = 'Aberdeen' AND T3.airline = 'United Airlines'` |

Table 4: An example error made by Codex with least-to-most prompting.

we show that generating sub-questions and the SQL query in a single pass can also achieve improved accuracy. Thus, iterative prompting, which is computationally costly, is not necessary when prompting LLMs to reason for text-to-SQL parsing.

Another interesting finding is that chain-of-thought prompting performs even worse than the standard prompting method. We analyze the reason in the next section, which helps answer *RQ2*.

## 5.2 Error Analysis

We conduct a quantitative error analysis of all four prompting methods with the component matching accuracy (Yu et al., 2018) on the Spider development set. Component matching accuracy is a fine-grained exact match metric that evaluates five SQL components, including `SELECT` clauses, `WHERE` clauses, `GROUP BY` clauses, `ORDER BY` clauses, and `KEYWORDS` (all SQL keywords, operators, and column names). Since exact match is too strict, we

also consider a component to be correct if the whole SQL query's test-suite accuracy is 1.

As shown in Table 2, our QDecomp and QDecomp+InterCOL prompts achieve better performance than other CoT-style prompting methods across all five SQL components. Further analysis shows that chain-of-thought prompting underperforms standard prompting because it provides very detailed reasoning steps. Translating such detailed steps is error-prone and incurs more error propagation issues. For example, in Table 3, Codex follows its reasoning steps faithfully to generate the corresponding SQL query, but the reasoning steps themselves have several errors, such as choosing the "breed_name" column instead of the "name" column in the `SELECT` clause. Least-to-most prompting makes improvements by providing reasoning steps at a higher level (via the problem reduction phase). However, it sometimes still cannot translate a sub-question into the correct SQL

| Selection Method | Spider Dev | | | | |
|---|---|---|---|---|---|
| | Easy | Medium | Hard | Extra Hard | Overall |
| Random | 89.6 | 74.1 | 52.4 | 38.1 | 68.4 ± 2.05 |
| G1 | **89.8** | **75.6** | 51.7 | 38.8 | **69.0** ± 2.18 |
| G2 | 87.4 | 72.2 | 50.4 | 39.4 | 66.9 ± 2.31 |
| G3 | 88.7 | 71.1 | **56.8** | **45.7** | 68.8 ± 1.16 |

Table 5: 8-shot test-suite accuracy of Codex on Spider Dev using **QDecomp+InterCOL prompting** with different in-context example selection methods.

| | Random | G1 | G2 | G3 |
|---|---|---|---|---|
| Standard | 63.2 | 64.1 | 60.2 | 58.2 |
| Least-to-Most | 65.8 | 62.6 | 61.2 | 63.3 |
| QDecomp | 67.4 | 68.2 | 65.2 | 66.6 |
| + InterCOL | **68.4** | **69.0** | **66.9** | **68.8** |

Table 6: 8-shot test-suite accuracy of Codex on the Spider dev set using different in-context example selection methods.

| | 0 | 1 | 4 | 8 |
|---|---|---|---|---|
| Standard | 59.6 | 62.0 | 63.9 | 63.2 |
| Least-to-Most | - | 59.2 | 62.1 | 65.8 |
| QDecomp | - | **63.1** | **66.6** | 67.4 |
| + InterCOL | - | 61.4 | 66.5 | **68.4** |

Table 7: Test-suite accuracy of Codex on the Spider dev set using different numbers of in-context examples. We do not have 0-shot results for the proposed methods as they need at least one example to learn how to solve the task step by step.

query, especially when involving hard components, such as `JOIN` clauses, `GROUP BY` clauses, and `ORDER BY` clauses (Table 2). We include an error example in Table 4. As a result, the errors are propagated to subsequent reasoning steps, leading to an incorrect final SQL parse. In contrast, QDecomp+InterCOL prompting outperforms these two methods because it does not instruct Codex to generate detailed reasoning steps or intermediate SQL queries. In this way, it reduces the possibility of accumulating mistakes in reasoning steps.

### 5.3 Robustness to Prompt Design

To further validate our conclusions in the main experiments, we conduct additional experiments to test the robustness of all four prompting methods in this section. Because chain-of-thought prompting already under-performs the standard prompting without reasoning, we omit this method in this and the next section.

**Selection of In-Context Examples.** Besides random selection, we evaluate the efficacy of QDecomp+InterCOL and other prompting methods with in-context examples at various difficulty levels. As Table 5 suggests, QDecomp+InterCOL enables Codex to learn to reason for SQL queries at different difficulty levels from in-context examples. When using G1 examples, Codex learns to generate SQL queries and reasoning steps of various lengths from G1 examples. Thus, it is less likely to generate redundant SQL clauses or reasoning steps and achieves the highest accuracy for SQL queries at easy and medium level. When using G3 examples,

Codex obtains the best performance for hard and extra-hard SQL queries. We conjecture that QDecomp+InterCOL teaches Codex to become better at generating SQL queries at difficulty levels similar to the in-context examples.

Base on the conjecture, we extend this experiment to compare QDecomp+InterCOL and other prompting methods. As shown in Table 6, QDecomp+InterCOL prompting achieves the best performance across all settings, demonstrating its robustness. However, least-to-most prompting does not benefit from G1 or G3 examples and shows decreased accuracy. We believe this performance drop is because its iterative prompting generates one reasoning step at a time, which is relatively independent of the overall reasoning step length.

**Number of In-Context Examples.** Intuitively, performances of all prompting methods improve as the number of in-context examples increases (Table 7). We found that our QDecomp prompting is the most robust and consistently achieves better performance than standard prompting. However, least-to-most prompting underperforms standard prompting when the number of examples is less than 8. In addition, we note that in our preliminary experiments, further increasing the number of examples only leads to minor gains. Hence, we use 8 in-context examples in our main experiments.

**Format of In-Context Examples.** Finally, we show the performance of Codex using two prompt formats, API docs and Create Table + Select 3

| | API docs | Create Table + Select 3 |
|---|---|---|
| Standard | 63.9 | 64.1 |
| Least-to-Most | 62.1 | 63.8 |
| QDecomp | **66.6** | **66.2** |
| + InterCOL | 66.5 | 64.3 |

Table 8: 4-shot test-suite accuracy of Codex on the Spider dev set using different prompt formats.

| | GeoQuery | IMDB | Yelp | MacroAvg |
|---|---|---|---|---|
| Standard | 60.99 | 73.28 | 45.31 | 59.86 |
| Least-to-Most | 60.99 | 58.78 | 36.72 | 52.16 |
| QDecomp | 64.84 | **77.86** | 48.44 | 63.71 |
| + InterCOL | **75.82** | 73.28 | **49.22** | **66.11** |

Table 9: 4-shot test-suite accuracy of Codex on three other text-to-SQL datasets across different prompting methods.

(Table 8). Due to OpenAI's prompt length restrictions, we use 4 in-context examples in this experiment. Although Create Table + Select 3 format includes foreign key information and database content, compared with API docs, it brings a negligible improvement in performance for standard prompting and a (slight) decrease for QDecomp and QDecomp+InterCOL prompting methods. Nonetheless, QDecomp is still the best prompting method under this format. Therefore, we use API docs as our default format due to its efficiency and leave further experiments for future work.

### 5.4 Results on Other Text-to-SQL Datasets

Besides the Spider datasets, we further compare QDecomp (+InterCOL) to standard and least-to-most prompting on other datasets including GeoQuery (Zelle and Mooney, 1996; Iyer et al., 2017), IMDB (Yaghmazadeh et al., 2017), and Yelp (Yaghmazadeh et al., 2017). Since the database schema and SQL queries in these datasets are more complex than those in the Spider datasets, we also use 4-shot in-context examples in this experiment.

As shown in Table 9, QDecomp (+InterCOL) consistently achieves the best performance for all three datasets. Moreover, we observe that least-to-most prompting underperforms standard prompting on IMDB and Yelp, which may be related to both iterative prompting and error propagation (Section 5.2). For example, least-to-most prompting would decompose the question "Find all movies that were produced by Netflix" into two sub-questions: 1) "Find all movies" and 2) "Find all movies that were produced by Netflix." Then, in the iterative solving stage, there are many correct SQL queries using different tables and columns for the first sub-question. Without seeing the second sub-question, it is hard for the LLM to pinpoint the correct ones. As a result, the LLM would include redundant or wrong schema items in the SQL parse for the first sub-question, which are propagated to subsequent steps. Since QDecomp (+InterCOL) instructs the LLM to generate the SQL query after all sub-questions are

derived, it maintains a global view of all reasoning steps and mitigates such error propagation issues.

## 6 Conclusion and Future Work

In this paper, we systematically explore CoT-style prompting to enhance LLMs' reasoning capability for text-to-SQL parsing. We design reasoning steps in order to apply two existing methods, chain-of-thought and least-to-most prompting, and propose new question decomposition prompting methods. Through comprehensive experiments, we demonstrate: (1) Iterative prompting may be not necessary for reasoning in text-to-SQL parsing. (2) Using detailed reasoning steps (in CoT) or intermediate SQL queries (in least-to-most prompting) is error-prone and aggravates error propagation.

Our question decomposition prompting serves as one of the first attempts to mitigate the error propagation issue in LLMs' multi-step reasoning, and we highlight this problem as a meaningful future direction. For example, we can further reduce errors in intermediate reasoning steps by incorporating our method into an interactive semantic parsing framework (Yao et al., 2019, 2020; Li et al., 2020; Zeng et al., 2020; Chen et al., 2023a,b). Since the decomposed sub-questions are in natural language, this interactive approach enables database users to easily spot the errors in each sub-question. Then, they can collaborate with LLMs by editing the sub-questions directly or providing natural language feedback (Elgohary et al., 2020, 2021; Narechania et al., 2021; Mo et al., 2022), which should further improve text-to-SQL parsing accuracy.

## Limitations

**Experiments on other large language models.** Our study focused on conducting experiments using Codex as the LLM, since it was available at no cost and showed impressive performance in text-to-SQL parsing among LLMs before GPT-4 (Rajkumar et al., 2022). To gain a comprehensive understanding of different CoT-style promptings for text-to-SQL, future research should explore the

effects of these promptings on more recent, more powerful LLM models, such as GPT-4 (if budget allows). By doing so, we can determine whether the improvements achieved by our proposed promptings are consistent across different LLMs.

**Experiments on robustness.** In our work, we mainly test robustness from the prompt design perspective such as how to select in-context examples, the number of in-context examples and the prompt format of in-context examples. It would also be valuable to investigate our prompting methods under different databases, natural language questions, or SQL perturbations (Chang et al., 2023a). This broader exploration would enable us to evaluate the robustness of our prompting methods across diverse scenarios.

## Acknowledgements

We would like to thank the anonymous reviewers and colleagues from the OSU NLP group for their thoughtful comments. This research was sponsored in part by a sponsored research award by Cisco Research, NSF IIS-1815674, NSF CAREER #1942980, NSF OAC-2112606, and Ohio Supercomputer Center (Center, 1987). The views and conclusions contained herein are those of the authors and should not be interpreted as representing the official policies, either expressed or implied, of the U.S. government. The U.S. Government is authorized to reproduce and distribute reprints for Government purposes notwithstanding any copyright notice herein.

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

## A  Example Prompts

```
### SQLite SQL tables, with their properties:
#
# medicine (id, name, trade_name, fda_approved)
# enzyme (id, name, location, product, chromosome, omim, porphyria)
# medicine_enzyme_interaction (enzyme_id, medicine_id, interaction_type)
#
```

Figure 2: An example for API docs prompt format, introduced by Rajkumar et al. (2022), on Spider.

```
CREATE TABLE grapes (
ID INTEGER PRIMARY KEY,
Grape TEXT UNIQUE,
Color TEXT
)
/*
3 example rows:
SELECT * FROM grapes LIMIT 3;
ID Grape Color
1 Barbera Red
2 Cabernet Franc Red
3 Cabernet Sauvingnon Red
/

CREATE TABLE appellations (
No INTEGER PRIMARY KEY,
Appelation TEXT UNIQUE,
County TEXT,
State TEXT,
Area TEXT,
isAVA TEXT
)
/*
3 example rows:
SELECT * FROM appellations LIMIT 3;
No Appelation County State Area isAVA
1 Alexander Valley Sonoma California North Coast Yes
2 Amador County Amador California Sierra Foothills No
3 Amador-Mendocino-Sonoma Counties N/A California N/A No
/

CREATE TABLE wine (
No INTEGER,
Grape TEXT,
Winery TEXT,
Appelation TEXT,
State TEXT,
Name TEXT,
Year INTEGER,
Price INTEGER,
Score INTEGER,
Cases INTEGER,
Drink TEXT,
FOREIGN KEY (Grape) REFERENCES grapes(Grape),
FOREIGN KEY (Appelation) REFERENCES appellations(Appelation)
)
/*
3 example rows:
SELECT * FROM wine LIMIT 3;
No Grape Winery Appelation State Name Year Price Score Cases Drink
1 Zinfandel Robert Biale St. Helena California Old Kraft Vineyard 2008 44 93 275
now
2 Zinfandel Chiarello Family Napa Valley California Giana 2008 35 93 480 now
3 Zinfandel Robert Biale Napa Valley California Black Chicken 2008 40 91 2700 2012
/
```

Figure 3: An example for Create Table + Select 3 prompt format, introduced by Rajkumar et al. (2022), on Spider.

```
### SQLite SQL tables, with their properties:
#
# medicine (id, name, trade_name, fda_approved)
# enzyme (id, name, location, product, chromosome, omim, porphyria)
# medicine_enzyme_interaction (enzyme_id, medicine_id, interaction_type)
#
### What is the total count of enzymes?
SELECT count(*) FROM enzyme

### SQLite SQL tables, with their properties:
#
# buildings (id, name, city, height, stories, status)
# companies (id, name, headquarters, industry, sales_billion, profits_billion,
assets_billion, market_value_billion)
# office_locations (building_id, company_id, move_in_year)
#
### Show the industries shared by companies whose headquarters are "USA" and
companies whose headquarters are "China".
SELECT Industry FROM Companies WHERE Headquarters = "USA" INTERSECT SELECT Industry
FROM Companies WHERE Headquarters = "China"

### SQLite SQL tables, with their properties:
#
# stadium (stadium_id, location, name, capacity, highest, lowest, average)
# singer (singer_id, name, country, song_name, song_release_year, age, is_male)
# concert (concert_id, concert_name, theme, stadium_id, year)
# singer_in_concert (concert_id, singer_id)
#
### How many singers do we have?
```

Figure 4: An example prompt under the standard API docs prompting for 2-shot on Spider.

```
### SQLite SQL tables, with their properties:
#
# book_club (book_club_id, year, author_or_editor, book_title, publisher, category,
result)
# movie (movie_id, title, year, director, budget_million, gross_worldwide)
# culture_company (company_name, type, incorporated_in, group_equity_shareholding,
book_club_id, movie_id)
#
### List categories that have at least two books after year 1989.
# Let's think step by step

# This query chooses records from the Book_Club table, followed by a WHERE clause
that selects records where the year column is greater than 1989.  It then groups the
results by the category column.  It then filters the results where the count of each
category is greater than or equal to 2.  It then selects the category column.

# Thus, the answer for the question is:  List categories that have at least two
books after year 1989.
SELECT category FROM book_club WHERE YEAR > 1989 GROUP BY category HAVING count(*)
>= 2

### SQLite SQL tables, with their properties:
#
# stadium (stadium_id, location, name, capacity, highest, lowest, average)
# singer (singer_id, name, country, song_name, song_release_year, age, is_male)
# concert (concert_id, concert_name, theme, stadium_id, year)
# singer_in_concert (concert_id, singer_id)
#
### How many singers do we have?
```

Figure 5: An example prompt under chain-of-thought + API docs prompting for 1-shot on Spider.

```
### SQLite SQL tables, with their properties:
# class (class_code, crs_code, class_section, class_time, class_room, prof_num)
# course (crs_code, dept_code, crs_description, crs_credit)
# department (dept_code, dept_name, school_code, emp_num, dept_address)
# employee (emp_num, emp_lname, emp_initial, emp_jobcode, emp_hiredate, emp_dob)
# enroll (class_code, stu_num, enroll_grade)
# professor (emp_num, dept_code, prof_office, prof_extension, prof_high_degree)
# student (stu_num, stu_lname, stu_fname, stu_init, stu_dob, stu_hrs, stu_class,
stu_gpa, stu_transfer, dept_code, stu_phone, prof_num)
# To answer the question "Find the first names and offices of all instructors who
have taught some course and the course description and the department name.", we
need to know:  "Find the first names and offices of all instructors.", "Find the
first names and offices of all instructors who have taught some course.", "Find
the first names and offices of all instructors who have taught some course and the
course description.".

### SQLite SQL tables, with their properties:
# station (station_id, name, annual_entry_exit, annual_interchanges, total_-
passengers, location, main_services, number_of_platforms)
# train (train_id, name, time, service)
# train_station (train_id, station_id)
# To answer the question "Show all train names and times in stations in London in
descending order by train time.", we need to know:
```

Figure 6: An example prompt under least-to-most + API docs prompting (problem reduction) for 1-shot on Spider.

```
### SQLite SQL tables, with their properties:
# class (class_code, crs_code, class_section, class_time, class_room, prof_num)
# course (crs_code, dept_code, crs_description, crs_credit)
# department (dept_code, dept_name, school_code, emp_num, dept_address)
# employee (emp_num, emp_lname, emp_initial, emp_jobcode, emp_hiredate, emp_dob)
# enroll (class_code, stu_num, enroll_grade)
# professor (emp_num, dept_code, prof_office, prof_extension, prof_high_degree)
# student (stu_num, stu_lname, stu_fname, stu_init, stu_dob, stu_hrs, stu_class,
stu_gpa, stu_transfer, dept_code, stu_phone, prof_num)
#
Q: Find the first names and offices of all instructors.
A: SELECT T1.emp_fname , T2.prof_office FROM employee AS T1 JOIN professor AS T2 ON
T1.emp_num = T2.emp_num

Q: Find the first names and offices of all instructors who have taught some course.
A: SELECT T2.emp_fname , T4.prof_office FROM CLASS AS T1 JOIN employee AS T2
ON T1.prof_num = T2.emp_num JOIN course AS T3 ON T1.crs_code = T3.crs_code JOIN
professor AS T4 ON T2.emp_num = T4.emp_num

Q: Find the first names and offices of all instructors who have taught some course
and the course description.
A: SELECT T2.emp_fname , T4.prof_office , T3.crs_description FROM CLASS AS T1
JOIN employee AS T2 ON T1.prof_num = T2.emp_num JOIN course AS T3 ON T1.crs_code
= T3.crs_code JOIN professor AS T4 ON T2.emp_num = T4.emp_num

Q: Find the first names and offices of all instructors who have taught some course
and the course description and the department name.
A: SELECT T2.emp_fname , T4.prof_office , T3.crs_description , T5.dept_name FROM
CLASS AS T1 JOIN employee AS T2 ON T1.prof_num = T2.emp_num JOIN course AS T3 ON
T1.crs_code = T3.crs_code JOIN professor AS T4 ON T2.emp_num = T4.emp_num JOIN
department AS T5 ON T4.dept_code = T5.dept_code

### SQLite SQL tables, with their properties:
# station (station_id, name, annual_entry_exit, annual_interchanges, total_-
passengers, location, main_services, number_of_platforms)
# train (train_id, name, time, service)
# train_station (train_id, station_id)
#
Q: Show all train names and times.
```

Figure 7: An example prompt under least-to-most + API docs prompting (problem solving) for 1-shot on Spider. The same prompt will be used to solve the next sub-question after we get the generated SQL query for the first sub-question.

```
### SQLite SQL tables, with their properties:
# document_types (document_type_code, document_description)
# documents (document_id, document_type_code, grant_id, sent_date, response_-
received_date, other_details)
# grants (grant_id, organisation_id, grant_amount, grant_start_date, grant_end_date,
other_details)
# organisation_types (organisation_type, organisation_type_description)
# organisations (organisation_id, organisation_type, organisation_details)
# project_outcomes (project_id, outcome_code, outcome_details)
# project_staff (staff_id, project_id, role_code, date_from, date_to, other_details)
# projects (project_id, organisation_id, project_details)
# research_outcomes (outcome_code, outcome_description)
# research_staff (staff_id, employer_organisation_id, staff_details)
# staff_roles (role_code, role_description)
# tasks (task_id, project_id, task_details, eg agree objectives)
#
### Question:  Find out the send dates of the documents with the grant amount of
more than 5000 were granted by organisation type described as "Research".
decompose the question

1.  Find out the send dates of the documents.
2.  Find out the send dates of the documents with the grant amount of more than
5000.
3.  Find out the send dates of the documents with the grant amount of more than 5000
were granted by organisation type described as "Research".

# Thus, the answer for the question is:  Find out the send dates of the documents
with the grant amount of more than 5000 were granted by organisation type described
as "Research".
SELECT T1.sent_date FROM documents AS T1 JOIN Grants AS T2 ON T1.grant_id =
T2.grant_id JOIN Organisations AS T3 ON T2.organisation_id = T3.organisation_id
JOIN organisation_Types AS T4 ON T3.organisation_type = T4.organisation_type WHERE
T2.grant_amount > 5000 AND T4.organisation_type_description = 'Research'

### SQLite SQL tables, with their properties:
# stadium (stadium_id, location, name, capacity, highest, lowest, average)
# singer (singer_id, name, country, song_name, song_release_year, age, is_male)
# concert (concert_id, concert_name, theme, stadium_id, year)
# singer_in_concert (concert_id, singer_id)
#
### Question:  How many singers do we have?
decompose the question
```

Figure 8: An example prompt under QDecomp + API docs prompting for 1-shot on Spider.

```
### SQLite SQL tables, with their properties:
# document_types (document_type_code, document_description)
# documents (document_id, document_type_code, grant_id, sent_date, response_-
received_date, other_details)
# grants (grant_id, organisation_id, grant_amount, grant_start_date, grant_end_date,
other_details)
# organisation_types (organisation_type, organisation_type_description)
# organisations (organisation_id, organisation_type, organisation_details)
# project_outcomes (project_id, outcome_code, outcome_details)
# project_staff (staff_id, project_id, role_code, date_from, date_to, other_details)
# projects (project_id, organisation_id, project_details)
# research_outcomes (outcome_code, outcome_description)
# research_staff (staff_id, employer_organisation_id, staff_details)
# staff_roles (role_code, role_description)
# tasks (task_id, project_id, task_details, eg agree objectives)
#
### Question:  Find out the send dates of the documents with the grant amount of
more than 5000 were granted by organisation type described as "Research".
decompose the question

1.  Find out the send dates of the documents.
SQL table (column):  documents (sent_date)
2.  Find out the send dates of the documents with the grant amount of more than
5000.
SQL table (column):  grants (grant_amount, grant_id)
3.  Find out the send dates of the documents with the grant amount of more than 5000
were granted by organisation type described as "Research".
SQL table (column):  organisation_Types (organisation_type_description,
organisation_type), organisations (organisation_type, organisation_id)

# Thus, the answer for the question is:  Find out the send dates of the documents
with the grant amount of more than 5000 were granted by organisation type described
as "Research".
SELECT T1.sent_date FROM documents AS T1 JOIN Grants AS T2 ON T1.grant_id =
T2.grant_id JOIN Organisations AS T3 ON T2.organisation_id = T3.organisation_id
JOIN organisation_Types AS T4 ON T3.organisation_type = T4.organisation_type WHERE
T2.grant_amount > 5000 AND T4.organisation_type_description = 'Research'

### SQLite SQL tables, with their properties:
# stadium (stadium_id, location, name, capacity, highest, lowest, average)
# singer (singer_id, name, country, song_name, song_release_year, age, is_male)
# concert (concert_id, concert_name, theme, stadium_id, year)
# singer_in_concert (concert_id, singer_id)
#
### Question:  How many singers do we have?
decompose the question
```

Figure 9: An example prompt under QDecomp+InterCOL + API docs prompting for 1-shot on Spider.