# OpenReview forum: "Exploring Chain of Thought Style Prompting for Text-to-SQL"
_EMNLP/2023/Conference — EMNLP 2023 Main_

### Official Review · Reviewer_vQpJ · 2023-08-03

**Soundness:** 4

**Excitement:**

4: Strong: This paper deepens the understanding of some phenomenon or lowers the barriers to an existing research direction.

**Paper Topic And Main Contributions:**

This paper explores enhancing the multi-step reasoning ability of LLMs in the context of text-to-SQL parsing. Specifically, this work applies CoT and least-to-most prompting to text-to-SQL parsing. In doing so, it finds issues of error propagation associated with these two prompting methods and highlights the unnecessity of iterative prompting. In response to these observations, the authors introduce a new CoT-style prompting method termed QDecomp, alongside its variant, QDecomp+InterCOL. The empirical evidence presented supports the enhanced accuracy of the proposed question decomposition prompting method. The experimental section offers a thorough analysis, shedding further light on the findings.

**Reasons To Accept:**

- The paper studies CoT style prompting within the domain of text-to-SQL parsing. The experimental findings provide two primary insights: (1) error propagation is introduced by these two methods frequently, and (2) iterative prompting may not be necessary when prompting LLMs for text-to-SQL parsing.
- The proposed question decomposition prompting method demonstrates an improvement in accuracy for the text-to-SQL parsing task compared to both the CoT and least-to-most prompting strategies.
- It is good to see experiments and analysis on the number, format, and selection strategy of in-context examples, which is useful for subsequent studies on designing prompting strategies for text-to-SQL parsing.
- The attempt to address and mitigate the error propagation of CoT prompting in the context of text-to-SQL parsing is interesting.

**Reasons To Reject:**

- Given the significant performance disparity observed between CoT-style prompting methods presented in this work and both fine-tuned methods [1] and other LLM-based methods [2], it is a lack of a detailed comparison providing insights into the performance differentials among the three.

[1] Qi, Jiexing, et al. "RASAT: Integrating Relational Structures into Pretrained Seq2Seq Model for Text-to-SQL." Proceedings of the 2022 Conference on Empirical Methods in Natural Language Processing. 2022.

[2] Ni, Ansong, et al. "Lever: Learning to verify language-to-code generation with execution." International Conference on Machine Learning. PMLR, 2023.

**Reproducibility:**

4: Could mostly reproduce the results, but there may be some variation because of sample variance or minor variations in their interpretation of the protocol or method.

**Reviewer Confidence:**

3: Pretty sure, but there's a chance I missed something. Although I have a good feel for this area in general, I did not carefully check the paper's details, e.g., the math, experimental design, or novelty.

---

> ### Author Rebuttal · Authors · 2023-08-29
>
> We greatly appreciate that the reviewer finds our method shows “an improvement in accuracy,” our analysis “useful for subsequent studies,” and our attempt to mitigate error propagation in CoT prompting “interesting.” To address the reviewer’s concern, we conduct a detailed comparison between our method and “fine-tuned methods” or “other LLM-based methods.” We will also add related discussions in our revised paper.
>
> "Comparison with Fine-Tuned Methods"
>
> In our new experiments (response to Reviewer DzXr), we show that harder in-context examples can further improve our methods’ accuracy to an average of 78.2 execution accuracy and a maximum of 79.8 among the 10 runs. We think that our method can achieve comparable performance to RASAT+PICARD (79.8 v.s. 80.5 on the Spider development set) [1], a method fine-tuned with supervised learning. Meanwhile, we would like to gently remind the reviewer that our method is few-shot in-context learning. Thus, direct comparison with fine-tuned methods using the full Spider training set may be unfair. Reducing the number of annotated text-to-SQL data is also one of the motivations of our work.
>
> "Comparison with other LLM-based Methods"
>
> We compare our method with other LLM-based methods [2][3]. As previously mentioned, the best execution accuracy our method can achieve is 79.8, when using hard in-context examples. Compared to [2], our method is 1.9 points lower than their execution accuracy (81.2). We hypothesize this is because [2] uses the execution results of SQL queries, which provides extra information for better database grounding. We would like to explore the use of database contents beyond table and column names in future work. Meanwhile, we note that LEVER’s verifier model is also fine-tuned on the full Spider training set, which may have extra advantages over our method. Our work is closest to DIN-SQL [3], which introduces a multi-stage prompting method for text-to-SQL parsing, and we can achieve better execution accuracy on the Spider development set (79.8 v.s 74.2; Tables 4 and 5 in [3]).
>
> References:
>
> [1] Qi, Jiexing, et al. "RASAT: Integrating Relational Structures into Pretrained Seq2Seq Model for Text-to-SQL." Proceedings of the 2022 Conference on Empirical Methods in Natural Language Processing. 2022.
>
> [2] Ni, Ansong, et al. "Lever: Learning to verify language-to-code generation with execution." International Conference on Machine Learning. 2023.
>
> [3] Pourreza, Mohammadreza and Rafiei, Davood. “DIN-SQL: Decomposed In-Context Learning of Text-to-SQL with Self-Correction.” Arxiv Preprint. 2023.

---

### Official Review · Reviewer_Cceb · 2023-08-03

**Soundness:** 3

**Excitement:**

3: Ambivalent: It has merits (e.g., it reports state-of-the-art results, the idea is nice), but there are key weaknesses (e.g., it describes incremental work), and it can significantly benefit from another round of revision. However, I won't object to accepting it if my co-reviewers champion it.

**Paper Topic And Main Contributions:**

The authors propose a CoT-style prompting method that improves LLMs’ multi-step reasoning for text-to-SQL task. They design a variant of question decomposition prompting by incrementally including the table and column names involved in each sub-question.

**Reasons To Accept:**

Propose a question-decomposition prompt.

**Reasons To Reject:**

- The result of experiment is not very satisfactory.
- Experimental results are fluctuate too much
- It is difficult to prove the validity and robustness of the method.

**Reproducibility:**

4: Could mostly reproduce the results, but there may be some variation because of sample variance or minor variations in their interpretation of the protocol or method.

**Reviewer Confidence:**

3: Pretty sure, but there's a chance I missed something. Although I have a good feel for this area in general, I did not carefully check the paper's details, e.g., the math, experimental design, or novelty.

---

> ### Author Rebuttal · Authors · 2023-08-29
>
> We thank the reviewer for the time and effort in reviewing our paper. On the other hand, there might be some misunderstanding and please let us clarify as follows:
>
> We want to first clarify that our paper's main contribution is not to “propose a CoT-style prompting method.” Instead, our work focuses on a novel exploration of diverse CoT prompting methods for text-to-SQL parsing. Specifically, we aim to tackle the two research questions identified in the paper (Lines 73-77). Not only do we adapt chain-of-thought prompting and least-to-most prompting to text-to-SQL parsing (which is the first effort for this task), but also propose a new question decomposition prompting method (QDecomp) and its variant (QDecomp + InterCOL). Hence, our work is not to “design a variant of question decomposition prompting” but to propose both QDecomp and its variant. With various experiments around the prompting methods, we offer valuable insights for the future development of effective prompting strategies for text-to-SQL parsing.
>
> “It is difficult to prove the validity and robustness of the method.”
>
> We’d appreciate it if the reviewer could elaborate on this concern, but let us highlight three aspects of our experiments, which aim to establish the validity and robustness of our proposed method. First, in our main experiments (Section 5.1), we conduct the evaluation on both the Spider development set and the Spider Realistic dataset. Spider Realistic removes or paraphrases direct mentions of table and column names in the original Spider questions, which shows our methods’ robustness against language variations. Second, we undertook a series of experiments in Section 5.3 to test our methods’ robustness against three factors of prompting, including the number, selection, and format of in-context examples. Finally, we show that our methods are also effective on three other in-domain datasets (Section 5.4), GeoQuery, IMDB, and Yelp, to provide further evidence of the effectiveness and resilience of our methods.
>
> “The result of experiment is not very satisfactory.” and “Experimental results are fluctuate too much.”
>
> We would greatly appreciate further details from the reviewer to elaborate on these comments. Currently, we are afraid that we don’t have a precise understanding of what the reviewer is concerned about. However, we attempt to explain as below:
>
> For the first comment, we firmly believe that our obtained results effectively address the two identified research questions (Lines 73-77). Our findings contribute substantially to these inquiries, and we are eager to engage in a constructive discussion with the reviewer to ensure that this perspective is well understood.
>
> For the comment “Experimental results are fluctuate too much”, we are uncertain which aspect the reviewer is referring to. Was the concern about the standard deviations in Table 1 being too high? If so, we believe this is largely due to the random selection of in-context examples. As Table 2 in [1] shows, the standard deviation on other NLP tasks can be as large as 14.7 when randomly selecting examples, not to mention that text-to-SQL parsing is a complex task that can result in large variances even for fine-tuned models (Table 3 in [2]).
>
> Your insights will greatly assist us in better understanding and addressing these concerns.
>
> References:
>
> [1] Zhang, Yiming, et al. “Active Example Selection for In-Context Learning.” Proceedings of the 2022 Conference on Empirical Methods in Natural Language Processing. 2022.
>
> [2] Deng, Xiang, et al. “Structure-Grounded Pretraining for Text-to-SQL.” Proceedings of the 2021 Conference of the North American Chapter of the Association for Computational Linguistics: Human Language Technologies. 2021.

---

### Official Review · Reviewer_DzXr · 2023-08-04

**Soundness:** 3

**Excitement:**

2: Mediocre: This paper makes marginal contributions (vs non-contemporaneous work), so I would rather not see it in the conference.

**Paper Topic And Main Contributions:**

In this work, the authors studied how to enhance language models' reasoning ability through CoT-style prompting.
CoT is enhanced by 1) QDecomp which decompose the complex query into simple ones and 2) InterCOL, incrementally including new data lines/rows.

This strategy is shown to be able to prevent error propagation and improve performances.



**Questions For The Authors:**

Have you conducted any significant tests and what are the results?

**Reasons To Accept:**

The propose method works well and the results show promising improvements.


**Reasons To Reject:**

The work is based on one db and it is not clear if it is applicable to other datasets. The guess is yes but cannot be verified.
Also it is not clear why  InterCOL's impacts are not consistent. It seems to work for some settings but has negative impacts under other settings. It is better to see results of InterCoL only.

**Reproducibility:**

3: Could reproduce the results with some difficulty. The settings of parameters are underspecified or subjectively determined; the training/evaluation data are not widely available.

**Reviewer Confidence:**

3: Pretty sure, but there's a chance I missed something. Although I have a good feel for this area in general, I did not carefully check the paper's details, e.g., the math, experimental design, or novelty.

---

> ### Author Rebuttal · Authors · 2023-08-29
>
> Thanks very much for your time reviewing our work! We are delighted that the reviewer finds our “proposed method works well” and “the results show promising improvements.” Here we address each of the reviewer’s concerns below and will add related discussion in our revised paper:
>
> “If it is applicable to other datasets”
>
> There might be some misunderstanding here. We agree with the reviewer that it is important to test how well our method generalizes across databases and datasets. To this end, we conducted our main experiments on Spider (Table 1), a text-to-SQL dataset with 200 databases. Moreover, Spider is a cross-database benchmark, which means that databases in its development set are not included in the training set. This evaluation setup exactly tests our method’s generalization across different databases. In addition to Spider, we also tested our method on three single-database datasets, GeoQuery, IMDB, and Yelp (Table 8). Thus, our work is not “based on one database” or one dataset.
>
> “Why InterCOL's impacts are not consistent”
>
> We find that the selection of in-context examples (Section 4.2) has the largest impact on QDecomp + InterCOL’s performance. Here we include some new experiments to show that when using harder in-context examples (denoted as G3, Lines 370-371), QDecomp + InterCOL consistently achieves the best performance in the main experiment and robustness experiments for the other two factors (number and format of in-context examples):
>
> |   Method  (Spider Dev)                    | Simple           | Medium           | Hard             | Extra Hard       | Overall TS (Overall EX)                                   |   |
> |-------------------------|------------------|------------------|------------------|------------------|-----------------------------------------------------------|---|
> | Standard (G3)           | 82.9             | 56.6             | 44.1             | 43.3             | 58.7 $\pm$ 3.32 (70.4 $\pm$ 3.68)                         |                                                   |
> | Least-to-Most (G3)      | 80.3            | 64.6             | 52.8             | 45.3             | 63.3 $\pm$ 1.95 (73.8 $\pm$ 1.72)                         |
> | QDecomp (G3)            | $\underline{87.9}$ | $\underline{67.5}$ | $\underline{54.1}$ | $\underline{45.1}$ | $\underline{66.5}$ $\pm$ 0.63 ($\underline{77.1}$ $\pm$ 0.75) |
> | + InterCOL (G3)         | $\textbf{88.7}$    | $\textbf{71.1}$    | $\textbf{56.8}$    | $\textbf{45.7}$    | $\textbf{68.8}$  $\pm$ 1.16 ($\textbf{78.2}$ $\pm$ 1.07)       |
>
> New Experiment 1: 8-shot test-suite (TS) and execution (EX) accuracy of Codex on the Spider development set using G3 examples and API Doc format.
>
> |               | API docs | Create Table + Select 3 |
> |:-------------:|:--------:|:-----------------------:|
> | Standard  (G3)    | 58.3     | 60.6                    |
> | Least-to-Most (G3) | 63.0     | 65.6                    |
> | QDecomp (G3)     | 65.5     | 66.9                    |
> | + interCOL (G3)   | 67.3     | 69.8                    |
>
> New Experiment 2: 4-shot test-suite (TS) accuracy of Codex on the Spider development set using G3 examples and two different formats.
>
>
> |               | 0    | 1    | 4    | 8    |
> |:-------------:|:----:|:----:|:----:|:----:|
> | Standard  (G3)    | 59.6 | 62.2 | 58.3 | 58.7 |
> | Least-to-Most (G3) | -    | 63.7 | 63.0 | 63.3 |
> | QDecomp (G3)      | -    | 63.6 | 65.5 | 66.5 |
> | + interCOL (G3)    | -    | 66.4 | 67.3 | 68.8 |
>
> New Experiment 3: Test-suite (TS) accuracy of Codex on the Spider development set with different numbers of in-context examples. We use G3 examples and API Doc format.
>
> We hypothesize that InterCOL benefits from harder in-context examples (G3) because they contain more complex table and column linking demonstrations. Consequently, QDecomp + InterCOL learn to ground table and column names to each subquestion better, which can guide Codex to generate the correct SQL query. On the other hand, with easier in-context examples:
>
> (1) [Table 5 in the paper] When the number of examples is insufficient (1 or 4), they may not provide clear guidance for Codex to accurately deduce the correct table-column pairs for harder questions at inference time. So, adding InterCOL to QDecomp may introduce more error propagation issues, which leads to a slight performance decrease.
>
> (2) [Table 7 in the paper] When we use the CREATE TABLE + SELECT 3 format, the rich schema information in this format can be more helpful than the easy demonstrations for table and column linking by InterCOL. So, QDecomp alone can achieve decent performance, and easy InterCOL examples cannot help as much as harder examples.
>
> Besides the new results, we would also like to kindly remind the reviewer that our work’s primary goal is to explore different CoT style prompting methods for text-to-SQL and try to address the two identified research questions (Lines 73-77), so proposing new CoT style prompting methods is only part of our contribution. Additionally, we propose both QDecomp and QDecomp + InterCOL, where the latter is a variant of QDecomp. In our paper, we have shown that QDecomp (+ InterCOL) consistently achieves the best performance across all experiments.
>
> We will add the new results and analysis in our appendix.
>
> “Have you conducted any significant tests and what are the results?”
>
> We perform McNemar’s Test on the median results out of the 10 runs in our main experiment (Table 1) to measure the statistical significance of the CoT-style prompting methods we have explored. Since chain-of-thought prompting performs worse than standard prompting, we focus on comparing the other three methods, least-to-most, QDecomp, and QDecomp + InterCOL, with standard prompting. By McNemar’s test, we find that all of the three methods achieve statistically significant improvement over standard prompting (p < 0.01). We will incorporate this result in our paper’s revised version.

---

### Meta-Review · Area_Chair_XDLg · 2023-09-11

**Recommendation:** 4

**Metareview:**

The paper verifies the chain-of-thoughts prompting style in the context of text-to-SQL. Two enhancements are introduced to the original CoT, providing interesting insights. The reviewers agree that the experimental results are promising. The authors have provided detailed answers to the questions raised by the reviewers (+ new results), addressing their concerns with respect to statistical significance of the results, the impact of different choices, and the novelty of the proposal. Reviewer DzXr has kept the initial judgement for the "possibility of missing or misunderstanding some points" while the concerns of reviewer Cceb are either not backed up with details or addressed in the rebuttal.

---

### Decision · Program_Chairs · 2023-10-07

**Decision:**

Accept-Main

**Comment:**

The paper verifies the chain-of-thoughts prompting style in the context of text-to-SQL. Two enhancements are introduced to the original CoT, providing interesting insights. The reviewers agree that the experimental results are promising. The authors have provided detailed answers to the questions raised by the reviewers (+ new results), addressing their concerns with respect to statistical significance of the results, the impact of different choices, and the novelty of the proposal. Reviewer DzXr has kept the initial judgement for the "possibility of missing or misunderstanding some points" while the concerns of reviewer Cceb are either not backed up with details or addressed in the rebuttal.